# Tick-Tock Consider the Clock: The Influence of Circadian and External Cycles on Time of Day Variation in the Human Metabolome—A Review

**DOI:** 10.3390/metabo11050328

**Published:** 2021-05-19

**Authors:** Thomas P. M. Hancox, Debra J. Skene, Robert Dallmann, Warwick B. Dunn

**Affiliations:** 1School of Biosciences, University of Birmingham, Edgbaston, Birmingham B15 2TT, UK; 2Chronobiology, Faculty of Health and Medical Sciences, University of Surrey, Guildford GU2 7XH, UK; d.skene@surrey.ac.uk; 3Division of Biomedical Sciences, Warwick Medical School, University of Warwick, Coventry CV4 7AL, UK; R.Dallmann@warwick.ac.uk; 4Institute of Metabolism and Systems Research, University of Birmingham, Edgbaston, Birmingham B15 2TT, UK

**Keywords:** circadian rhythms, diurnal rhythms, metabolomics, metabolite rhythms, blood, urine, saliva, breath, skeletal muscle

## Abstract

The past decade has seen a large influx of work investigating time of day variation in different human biofluid and tissue metabolomes. The driver of this daily variation can be endogenous circadian rhythms driven by the central and/or peripheral clocks, or exogenous diurnal rhythms driven by behavioural and environmental cycles, which manifest as regular 24 h cycles of metabolite concentrations. This review, of all published studies to date, establishes the extent of daily variation with regard to the number and identity of ‘rhythmic’ metabolites observed in blood, saliva, urine, breath, and skeletal muscle. The probable sources driving such variation, in addition to what metabolite classes are most susceptible in adhering to or uncoupling from such cycles is described in addition to a compiled list of common rhythmic metabolites. The reviewed studies show that the metabolome undergoes significant time of day variation, primarily observed for amino acids and multiple lipid classes. Such 24 h rhythms, driven by various factors discussed herein, are an additional source of intra/inter-individual variation and are thus highly pertinent to all studies applying untargeted and targeted metabolomics platforms, particularly for the construction of biomarker panels. The potential implications are discussed alongside proposed minimum reporting criteria suggested to acknowledge time of day variation as a potential influence of results and to facilitate improved reproducibility.

## 1. Introduction

A favoured application of metabolomics is that of metabolic phenotyping, typically for biomarker discovery and better understanding of disease pathology within the context of the functions of metabolites observed in human metabolomes [1,2], with such studies applied to the most prevalent chronic diseases within the human population such as cancer, diabetes and cardiovascular disease [3]. However, the concentration of metabolites in human biofluids and tissues is not static and varies across timescales of seconds to decades, driven by biological functions. Observed variation between such studies is inevitable due to biological variation within and between subjects as a result of genotype, environment, lifestyle, and the influence of biological rhythms, some of this variation being mitigated by control and monitoring of participant behaviour/diet/environment before and/or during the study [4]. In addition to biological variation, there is also analytical variation, resulting from sample selection/preparation [4,5], sample degradation as a result of storage conditions and freeze-thaw cycles [6,7,8], instrument variation [9] and varied methods of data pre-treatment, statistical analysis and modelling [5,10,11]. Standard practices in the field such as quality control processes, batch correction, data normalisation, and matched cohorts mitigate or report some of this variation [1,11] but the extent and rigor in which such practices apply vary from study to study, all of which impact the analysis and resulting biological interpretation [12]. This contributes to the ‘reproducibility crisis’, i.e., a recurring inability for external groups to reproduce published results, culminating in numerous independent research teams achieving conflicting results or inadvertently promoting false-positive findings stemming from replicating effects and bias [10,13,14,15]. Considine [10] puts forward an informed and comprehensive picture of the challenges of reproducibility and the implications for metabolic profiling and biomarker discovery. However, a key variable which may explain some variation in the data within and between studies is repeatedly overlooked; that of circadian control and diurnal variation which regulate 24 h biological rhythms including the metabolome. This results in significant changes to observed concentrations of a range of metabolites and hormones dependent on the time of day, with a well-known example being cortisol [16], with proline and leucine being further examples of highly rhythmic metabolites. Studies discussing these and further rhythms in metabolite concentrations and the context of these observations are discussed below. A number of detailed reviews and tutorials for biomarker discovery have also overlooked circadian control and diurnal variation of the metabolome as a consideration or influencing factor in study design and sample collection [5,17,18,19,20,21,22], with others only acknowledging circadian control as influencing metabolism in passing [23,24] and fewer still suggesting that it be considered in study design [25].

The omission of time of day variation as an important feature in metabolomics study design likely results from the fact that work characterising circadian and diurnal influence through such platforms has almost exclusively occurred over the past decade. The growing body of circadian metabolomics research should be of interest to the metabolomics community due to identifying metabolites that exhibit significant 24 h rhythmic behaviour as well as detailing the context and conditions in which this variation is observed. Such findings may prove informative when constructing or interrogating a biomarker panel, dissecting sources of inter-individual variation within studies as well as opening an additional line of enquiry when conflicting results arise between studies. It is timely that this body of work is thoroughly reviewed to establish the current knowledge base and consider the implications with regard to reproducibility especially in the construction of biomarker panels.

Whilst an in depth knowledge of chronobiology is not required to appreciate the findings of the studies discussed within this review, a general understanding of biological rhythms, specifically circadian and diurnal rhythmicity, as well as internal biological clocks, their relationship with metabolism, and how they can be monitored will offer some insight into study design and the importance of context with regard to how results were collected. As such, a brief overview and further reading are detailed below.

### 1.1. Key Concepts of Circadian Biology

How circadian rhythms propagate and exert temporal control over physiological processes is reviewed elsewhere [26]. In brief, circadian rhythms are endogenous, approximately 24 h oscillations in biological processes; such rhythms are generated by internal clocks, which are present in almost every cell of the body. On the cellular level, the ‘clock machinery’ is comprised of a transcription–translation autoregulatory feedback loop of core ‘clock genes’, which generate rhythmic outputs, e.g., the rhythmic expression of clock-controlled genes (CCGs), in turn conferring rhythmicity to the transcriptome, proteome and ultimately the metabolome. By definition, circadian rhythms are temperature compensated, resulting in no change to the rate of circadian oscillations across a significant physiological range of temperatures [27]. Moreover, biological clocks have evolved to be entrained (synchronised) by external factors, referred to as Zeitgeber, such as light/dark or feeding/fasting cycles, thus aligning themselves to resonate (match) with the Earth’s natural daily and seasonal cycles. Notably, circadian rhythms persist in free-running conditions, i.e., in the absence of Zeitgeber, thereby demonstrating their endogenous origin. An extensive body of research in animal models has not only established the cellular molecular machinery of the clock, but also the interconnectivity of all clocks within an organism. The mammalian timing system is typically divided into a central pacemaker in the ventral hypothalamus, i.e., the suprachiasmatic nuclei (SCN), and peripheral clocks in all other tissues of the nervous system and the body. The former receives light input from the eyes and, through neuronal and humoural signals, interacts with the peripheral clocks; with the SCN portrayed as interacting with all other clocks in the body, much like a conductor of an orchestra [28]. Furthermore, animal models have also demonstrated a strong link between the central circadian system, metabolism, and metabolite rhythms. Peripheral clocks possess some degree of autonomy. However, maintenance of ‘full circadian function’ relies on inter-clock signals between peripheral clocks and the SCN [29]. As described above, the clock machinery is capable of regulating metabolic processes, yet in turn, metabolites as well as hormones are capable of feeding back and regulating the core clock machinery and interfacing between peripheral/central clocks [30]. This interface manifests as temporally correlated metabolic processes, reflected by temporally correlated metabolites, across various tissues as demonstrated in mice [31,32].

Perturbations to the metabolome, potentially brought about by nutrient challenge or time restricted feeding, can influence the metabolism–clock interface resulting in loss of temporal correlation of metabolites between tissues or alterations in metabolite rhythms and, in specific instances, altered rhythms of core clock genes [31,33]. Desynchrony between peripheral clocks (circadian misalignment), potentially invoked by aberrations in the metabolism–clock interface, has been linked to numerous chronic conditions and metabolic disorders, with circadian-controlled genes being enriched amongst disease-associated genes vs. non-circadian-controlled genes [34]. Such diseases associated with circadian misalignment include hormone-dependent cancer such as breast and prostate cancer in shift workers [35], coronary heart disease [34,36], neurological disorders and negative impacts on mental health and well-being [37], in addition to metabolic disorders such as diabetes [38], all of which have an increased prevalence in shift workers according to epidemiological evidence available [36,39]. Predisposition and progression of these conditions appear to be underpinned by key changes in metabolism driven by circadian misalignment, such as increased insulin resistance and perturbed glucose metabolism and energy expenditure being core risk factors for developing type 2 diabetes and/or cardiovascular disease [40]. The intrinsic link between metabolism and the circadian system, as demonstrated in animal models, and the epidemiological evidence emphasising the clinical relevance of this research has likely spurred on circadian studies in humans, employing metabolomics.

Human circadian rhythms can be identified via constant routine studies (gold-standard protocol), controlling and minimising the influence of Zeitgeber. Specifically, ambient conditions such as light, temperature, posture, activity, wakefulness, and diet of participants, which otherwise may mask circadian rhythms, are kept constant throughout this type of study. How a constant routine protocol is designed and typically employed is described in detail by Duffy & Dijk [41], with further details on the process of entrainment and the nature of un-entrained (free-running) rhythms provided in [42,43]. Human studies, such as those discussed below, typically employ small but highly controlled study groups to minimise intra- and inter-subject variation and restrict confounding factors which may otherwise mask the rhythms being observed.

The circadian rhythms observed in constant routine conditions will differ in their phase (timing) and amplitude between individuals, primarily as a result of their genetics, age, and sex. Even if samples are taken at the same social time, e.g., 08:00 h, participants will likely express variation in the phase of their individual circadian rhythm; the relationship between the timing of an individual’s circadian rhythm and the timing of a Zeitgeber being referred to as the phase angle of entrainment [44,45,46]. Much like the hands of an actual clock revealing the current time of day there are (bio)markers that reveal the circadian clock’s phase, i.e., internal (biological) time. Widely used bona fide central clock phase markers are salivary/plasma melatonin, urinary 6-sulfatoxymelatonin (aMT6s) and plasma cortisol [47,48,49]; they are thought to reflect the timing of the SCN clock of an individual. Biological rhythms with a period of 24 h that do not/are not known to meet the defining principles of circadian rhythms, as outlined above, may be referred to as diurnal rhythms, i.e., 24 h rhythms observable under entrained “real-life” conditions jointly driven by exogenous factors, e.g., light/dark or feeding/fasting cycles, and the endogenous circadian timing system. As a trivial example, consumption of xenobiotics such as caffeine could lead to a 24 h rhythm peaking in the afternoon, as caffeine consumption stereotypically occurs in the morning, and reaching a nadir overnight. However, such a rhythm would not persist under constant conditions (no caffeine consumption) and thus is not circadian but a diurnal, evoked rhythm. By contrast, 24 h melatonin and cortisol rhythms persist under constant routine conditions and are thus deemed circadian. The terms should be carefully employed to accurately reflect the conditions under which 24 h rhythms were observed; a ~24 h cycle is not necessarily circadian—a common misinterpretation in the literature. Example rhythms exemplifying classification as a circadian or diurnal rhythm are provided in Figure 1 and real data demonstrating the influence of entraining agents (e.g., meals, sleep/wakefulness) are provided in Figure 2.

Considering the broad context in which time of day variation can be observed, and the implications that this may have for reproducibility between studies, we review all original research to date on human participants that applied either an untargeted or targeted metabolomics platform to observe time of day variation. Specifically, we have the following objectives: Establish the tissues in which time of day variation of metabolites have been observed, or failed to be observed, and the extent to which the metabolome is influenced.Establish the source(s) for this observed daily variation and, if applicable, which metabolite classes are most susceptible.Consider the implications of circadian/diurnal variation and the timing of sample collection on biomarker discovery and how this may undermine their potential clinical application.

### 1.2. Literature Search—Parameters and Outcomes

Prior to starting the literature search, three database/search engines were selected based on differing breadth of literature curation and indexing features to facilitate specific searching. PubMed (NCBI) provides specific curation of biomedical literature alongside subject headings (indexing) in the form of Medical Subject Headings (MeSH) terms for improved specificity of returned literature from submitted searches. Web of Science was selected for offering wider curation of literature, which may potentially not be included on the MEDLINE database used by PubMed. However, Web of Science does not use subject headings which may reduce search specificity vs. PubMed but still allows for Boolean operators (e.g., and, or) to combine search terms. Lastly, Google Scholar was chosen due to a lack of specific curation and similar search tools as Web of Science. Combined together all three platforms were deemed more than adequate to return the majority of relevant literature available, with further manual searching performed thereafter. Search terms relating to the subject fields of interest were chosen and, where possible, matched to MeSH terms—it was assumed that MeSH terms such as metabolomics likely correspond to common key words associated with literature across databases. Various combinations of search terms were trialled to acquire a search result that was considered broad in terms of ‘hits’, i.e., tens to low hundreds, but not so broad as to be infeasible to read and evaluate. Yielded literature from the search was counted as relevant based on the article title, article keywords, or abstract broadly referring to metabolomics/metabolism and circadian rhythms/chronobiology. Relevant literature collected from the database search were subject to inclusion criteria to assess if they were capable of addressing the outlined objectives, as stated above. Inclusion criteria consisted of three simple parameters, applied in the following order:The literature details original research, i.e., no derivative work such as reviewsThe research studied human participants over a time courseEmployed any metabolomics platform to analyse samples collected across the time course.

After this assessment of relevant literature yielded from the database search, manual searching was performed. Manual searching consisted of reading literature that met inclusion criteria and checking reference lists which may refer to further relevant studies (based on title/keywords/abstract) which would then be subject to inclusion criteria. Further details on search parameters and outcomes are shown in Table 1.

## 2. Literature Search—Results and Commentary

### 2.1. Blood

For the purpose of brevity, details relating to study design are not listed within the main body of text unless relevant to the provided commentary. Instead, key aspects of study design alongside results pertinent to the outlined objectives are summarised in tables after each subsection. Subsections and summary tables appear in the following order: blood (plasma/serum), urine, saliva, breath, and skeletal muscle. Furthermore, supplemental material containing compiled lists of metabolites showing time of day variation, and the context in which this variation took place, has been produced (Appendix A) in addition to an overview of observed time of day changes across the studies (Appendix A).

Blood is the most investigated biofluid assessed for time of day variation of the metabolome, based on the outcome of the performed literature search, with 18 studies discussed here. The collated literature implicated various factors contributing to time of day variation (Table 2) and other findings of interest which are presented below as common subthemes that we will carry forward, where applicable, to subsequent sections for the other sample matrices.

#### 2.1.1. Circadian Variation

Dallmann et al. [56] was first to publish a constant routine study observing circadian control of the metabolome in plasma and saliva (latter discussed below—see Section 2.3.1), with 41 (15%) of identified metabolites within plasma displaying circadian rhythms, of which, >75% comprised lipids, with nearly all peaking at subjective lunch time (when participants would expect their second daily meal under a typical sleep/wake, feeding/fasting cycle). These findings provide evidence that lipid metabolism is under circadian control. The amplitude of these observed rhythms differs between metabolites, with lactate abundance varying by ~66% and glutamate by ~40% across the 24 h period as but two examples, with other observed metabolites displaying even greater variation. This is a pertinent finding considering active discussion in the literature for these two metabolites as potential biomarkers and therapeutic targets for various diseases [69,70,71,72,73,74,75] and a consideration that can be applied to the findings of all the papers discussed below. In a separate constant routine study published in the same year [57] it was reported that phenylalanine, tryptophan, and leucine were observed to display circadian rhythms. In total, eight rhythmic metabolites were observed within the study, a comparison to the 41 observed previously [56] was drawn and the conclusion reached was that the authors likely observed less rhythmic metabolites due to curated selection of features with minimal variability between participants. Chua et al. [53] observed a comparable level of circadian rhythmic metabolites (35/263 metabolites, 13.3%) in their lipidomics assay, with triacylglycerides (TAGs) and diacylglycerides (DAGs) peaking in the morning. However, of these metabolites, only 12 (5%) to 86 (33%) displayed rhythmicity, with a median of 20% across all participants. Importantly, there were also significant differences in the phase of the rhythmic metabolites between participants, some as great as 12 h. These phase differences were observed despite similar cortisol and melatonin rhythms between participants, metabolite rhythms typically measured relative to dim light melatonin onset (DLMO) or when cortisol levels peak to account for individual differences in circadian timing (chronotype). As age/sex/ethnicity were all controlled for, the authors concluded that the three observed ‘lipidome phenotypes’ may be a result of predetermined genetic differences. This is a reasonable assertion echoed by a prior review [76], suggesting that genetic variation in human clock genes could contribute to phenotypic differences and is supported by twin and familial studies stating 27–50% of the variance in diurnal preference/chronotype is attributable to genetic, not environmental, influence [77,78,79,80]. 

These findings suggest that a portion of the plasma metabolome/lipidome is under circadian influence (Appendix A), the extent of which is potentially confounded by inter-individual differences in diurnal preference/circadian timing (chronotype). 

#### 2.1.2. Sleep Deprivation and Prolonged Wakefulness 

The aforementioned studies employ a constant routine protocol enforcing wakefulness over the observed time period. In an interesting development, Davies et al. [51], proceeded by Chua et al. [59], Grant et al. [65], and Honma et al. [50], characterise the impact of wakefulness (total sleep deprivation) on the plasma metabolome (Appendix A). A distinction between these four studies is that both Chua et al., and Grant et al., performed their sample collection during a constant routine protocol where participants were subject to total sleep deprivation for the duration whilst Davies et al., and Honma et al., employed an entrained protocol with two and three phases, respectively, across consecutive days with a 8 h period of sleep permitted during the first 24 h of sampling proceeded by 24 h of total sleep deprivation with Honma et al., monitoring a third day permitting ‘recovery’ sleep. The experimental design of both Davies et al., and Honma et al., studies allowed for paired comparisons between test conditions (e.g., sleep vs. no sleep) for individual participants.

Of the 109/171 (64%) rhythmic metabolites observed by Davies et al. (amino acids, biogenic amines, lipid groups), 95 (87%) of which peaked between 06:00 and 18:00 h, 21 (28%) lost rhythmicity during prolonged wakefulness alongside reduced/increased amplitude of metabolites that peaked within the day/night cycle, respectively. Furthermore 27/171 (16%) metabolites (three sphingolipids, eight acylcarnitines, 13 glycerophospholipids, tryptophan, serotonin, taurine) exhibited a significant increase during sleep deprivation vs. baseline sleep with serotonin exhibiting the largest change (44% ± 20%), illustrating a perturbation to metabolite rhythms coinciding with total sleep deprivation. These results were echoed by Chua et al. [59] and Grant et al. [65] who observed linear changes in 25/11 (9.5%/11.1%) metabolites, respectively, predominantly phosphatidylcholines/TAGs in the former study (lipidomics assay) and amino acids in the latter (HILIC (hydrophilic interaction chromatography) assay), as a result of sleep deprivation. Further agreement between Davies et al., and Chua et al., was observed for diacyl-phosphatidylcholines, which were affected by sleep deprivation.

Having performed untargeted/targeted HILIC assays, as opposed to reversed-phase assays which were performed previously [51,53], Grant et al., observed the impact of prolonged wakefulness on polar metabolites, with 28/99 (28.3%) displaying rhythmic-rhythmic/linear metabolites peaking in the biological night, of which 13 were amino acids, a similar result with regard to amino acids rhythms having been observed to some degree previously [55,56,57,61]. Furthermore, Grant et al., identified nine novel rhythmic polar metabolites (organic acids, nucleotides, and an amino acid) not observed in prior non-polar/lipid studies. Group-level analysis of their untargeted dataset mostly reflected what was observed in the targeted dataset (see Table 2).

Honma et al. [50] performed a similar study to Davies et al. with the exception of an all-female cohort (Davies et al., studied an all-male cohort) and therefore offers a novel insight into sex-dependent differences resulting from prolonged wakefulness (see Appendix A). At face value, the female cohort were comparable to the male cohort of Davies et al., with 58/130 (44.6%) metabolites being rhythmic and common across the three study days (baseline sleep, prolonged wakefulness, recovery sleep), and 97/130 (75%) were rhythmic on at least one of the three days but not rhythmic across all three. During sleep deprivation 15/130 (12%) of metabolites in female participants were significantly altered, of which 14 decreased in concentration, in contrast to Davies et al., in which 37/141 (26%) were significantly increased during sleep deprivation. Furthermore, a subset of 32 common rhythmic metabolites between the cohorts of both studies was analysed with regard to their mean acrophase (peak time) and it was observed that the acrophase in the female group was ~1 h later compared to the male group (female = 15:48 ± 0:40 h, male = 14:53 ± 0:42 h). Whilst the datasets are from different studies, they were both performed by the Skene laboratory (University of Surrey) with highly similar methodology, the same commercial kit to perform the targeted assay, similar cohorts (excluding sex) and subject to the same data processing and analysis, making these observations more compelling. Complementary to the findings of Davies et al., and of Skene et al. [64] (discussed below—see Section 2.1.3), the findings of Honma et al., also illustrate how acute disruption of the sleep–wake cycle can result in changes to metabolite rhythms that persist after the disrupted behavioural cycle is restored.

#### 2.1.3. Shift Work

In work that runs parallel to the studies investigating prolonged wakefulness and sleep deprivation, Skene et al. [64] and Kervezee et al. [67] independently investigated and characterised the impact of circadian misalignment on the plasma metabolome brought about by simulated shift work (Appendix A). Skene et al. compared simulated day (DS) vs. night (NS) shift work and focused on which metabolite rhythms are primarily driven by the central SCN clock or external behavioural cycles.

In the constant routine period succeeding the simulated shift work, Skene et al. observed 65/132 (49.2%) rhythmic metabolites following one or both conditions, with a further 19 metabolites (seven amino acids, 12 lysophosphatidylcholines) losing rhythmicity following NS and a second independent group of 19 metabolites (mostly phosphatdylcholines, acylcarnitines) gaining rhythmicity only after NS. There were 27 common rhythmic metabolites between test groups, only three of which maintained the same peak time between DS and NS. This was also a common trait with the circadian markers melatonin DLMO and cortisol, indicative to the authors that it is likely that these three metabolites (taurine, serotonin, sarcosine (N-methylglycine)) are strongly influenced by timing of the central SCN clock. Conversely, the remaining 24 rhythmic metabolites (mostly glycerophospholipids and sphingolipids) exhibited a significant shift in peak time between test conditions, with the majority experiencing a 12 h delay, i.e., inversed rhythms, and thus suggested to be strongly influenced by the shifted behavioural cycles (e.g., sleep/wake; feeding/fasting). This delay was maintained in the constant routine following cessation of the shift conditions demonstrating that endogenous metabolite rhythms can be induced and driven by external behavioural cues which likely reflect peripheral oscillators dissociating from that of the SCN, as concluded by the authors.

Similar to Skene et al., Kervezee et al. [67] observed a change in rhythmic metabolites in NS, with 51/130 (39.2%), 53 (40.8%), and 32 (24.6%) metabolites having displayed rhythmicity at baseline, NS, and in both datasets, respectively. Of the 32 metabolites rhythmic in both conditions (baseline and NS), 24 (75%) exhibited a phase shift, on average of 8.8 h, matching the shift in sleep pattern. Therefore, these changes are deemed more strongly influenced by behavioural cycles as opposed to circadian control with amino acids being an enriched group within the 24 metabolites exhibiting a phase shift. A further seven (~22%) metabolites, from the subset of 32, remained aligned with the non-shifted melatonin phase (measure of SCN phase). Therefore, this subset of seven metabolites (Appendix A) were deemed circadian SCN clock regulated, or at least not regulated by the sleep/wake cycle. The 19 metabolites that lost rhythmicity post-shift work simulation were predominantly lipids, similar findings shown by Davies et al. [51] and Chua et al. [59] when investigating the impact of sleep deprivation and corroborating with the results of Skene et al., [64].

The observation that behaviourally induced rhythms can be retained and persist for at least 24 h in free-running constant routine conditions complements the findings of Davies et al. [51] and Honma et al. [50] thus shift work and atypical sleep patterns are important considerations for metabolomics studies prior to recruitment and sampling.

#### 2.1.4. 24 h Diurnal Rhythms

Parallel to studies employing constant routine methodologies to investigate circadian rhythms are those which investigate diurnal rhythms, 24 h rhythms under entrained conditions. Three further studies, not investigating shift work or the impact of sleep, were obtained from the performed literature search characterising time of day variation over >24 h time courses. Park et al. [54] identified 34 metabolites exhibiting time of day variation between three time classes (‘morning’, ‘afternoon’ and ‘night’) covering a time course of 24 h. These distinct time classes were produced by averaging hourly ^1^H-NMR (nuclear magnetic resonance) spectra across all 10 participants for the 25 time points and by performing PCA analysis. Identification of time of day variation was performed with a bespoke two sample t-test, with correction for multiple testing, performed to test for significant differences in spectral regions between the three ‘time classes’. Ang et al. [55] showed time of day variation across 24 h in a range of metabolite classes using untargeted LC-MS (liquid chromatography mass spectrometry). They considered their work as an external validation of the Dallmann et al. study [56], having replicated some of their findings despite some key differences in methodology (see Table 2). In total, 34 rhythmic metabolites were identified with variation in abundance ranging from 49–81% (average 65%). The magnitude of this variation was similar to that observed by Dallmann et al., with amino acids and phospholipids being two highlighted rhythmic metabolite classes that displayed similar amplitudes. Gu et al. [66] observed diurnal variation in three participants within their pilot study to perform individual-level analysis to better explore inter-individual variability as opposed to group-level analysis; with only Chua et al. [53,59] having observed individual rhythms and differences prior to this publication and Grant et al. [65] also performing individual-level analysis and publishing in a similar timeframe. Gu et al., observed diurnal rhythms in all three participants (Table 2, Appendix A) in addition to observing inter-individual differences in the acrophase of metabolite rhythms relative to DLMO, such differences may be masked in group-level analysis but are pertinent when investigating on the individual level.

#### 2.1.5. Health Status

It is well established that different physiological and pathological states caused by disease can lead to distinct metabolic profiles. Isherwood et al. [61] adds to this understanding by observing unique time of day variations accompanying specific phenotypes (Appendix A). Their study observed differences in metabolic profiles of individuals with type 2 diabetes mellitus (T2DM), overweight/obese (OW/OB) non-diabetic individuals, and age-matched lean ‘healthy’ controls; the novel aspect of the study being the comparison of metabolic profiles between non-diabetic/diabetic individuals (age- and weight-matched) across multiple time points about a 24 h time course. In total, 50/130 (~38.5%) unique metabolites displayed 24 h rhythmicity of which 35, 39, and 20 were observed in lean, OW/OB and T2DM groups, respectively, from group-level analysis (Appendix A). Of these 50 rhythmic metabolites, 5 (10%) were unique to OW/OB and T2DM groups and 11 (22%) were unique to non-T2DM groups suggesting a change in metabolite rhythms associated with the onset or continuation of T2DM (Appendix A).

In a similar line of enquiry to Isherwood et al., Gehrman et al. [62] investigated how diurnal rhythms may differ between participants confirmed to suffer with insomnia and matched healthy individuals (‘good sleepers’). Unique metabolite profiles were observed for both test groups (Appendix A) with 29 metabolites being elevated/decreased in insomnia patients in the morning and/or night. Of these 29 metabolites, 13 were rhythmic. In total, 11 metabolites exhibited diurnal rhythms in both groups with a further six and seven unique rhythmic metabolites in healthy and insomniac participants, respectively. Phase changes were also noted with a phase advance (peaking earlier in the day) of acetone, proline, and a phase delay (peaking later in the day) of lactate, valine, isoleucine and 3-methyl-2-oxovalerate for the insomniac participants. It remains unclear how many of these changes are associated with the underlying causes of insomnia as opposed to the symptoms, i.e., poor sleep quality/duration.

Lusczek et al. [68], investigated whether intensive care unit (ICU) patients have their 24 h rhythms disrupted relative to healthy controls by profiling “circadian” rhythms in vital signs and plasma metabolites by analysing 60 “circadian” metabolites based on the work by Dallmann et al. [56] and Ang et al. [55] (who observed diurnal not circadian rhythms) and concluded that ICU patients experience desynchrony, leading to a loss of metabolite rhythmicity when compared to healthy non-ICU controls. The rhythmic metabolites observed by Dallmann et al., and Ang et al., were observed in a different and specific context, and do not serve as adequate or reliable markers for circadian desynchrony in the context of the study by Lusczek et al. Further measures taken to assess patient “circadian” rhythms included temperature, heart rate and blood pressure, and plasma cortisol of which only cortisol was deemed significantly rhythmic for group-level analysis of the ICU group. Lusczek et al., suggests that this is indicative of a lack of coherence in circadian phases and amplitudes amongst the ICU patients. However, an alternative suggestion is that coherence of cortisol rhythms, is indicative of a similar phase angle of entrainment between participants and the variable temperature and blood pressure rhythms result from patients’ unique circumstances with regard to their disease status (cardiac or respiratory), physiology/trauma, concomitant medication and demographics [81,82], disease state and physiology potentially leading to unique rhythmic metabolite profiles not necessarily related to desynchrony as established above.

#### 2.1.6. Diet Composition

The previously described studies have all collected blood samples at 2–5 h intervals across a minimum 24 h time course to define and visualise detected metabolite rhythms over a 24 h period via cosinor analysis (applies the least squares method to fit a sine wave to time series data) or MetaCycle (runs three separate algorithms to detect biological rhythms). Some studies may opt to characterise time of day variation over a shorter time course (<24 h) or with fewer samples per 24 h cycle (*n* ≤ 4 samples) and thus investigate variation in a morning vs. evening fashion to discern ‘gradient’ metabolites (‘gradient’ referring to a significant relative increase/decrease in concentration/abundance of a metabolite between two time points) such as the work performed by Sato et al. [63]. To clarify, low resolution sampling of *n* ≤ 4 samples per 24 h cycle is insufficient to characterise 24 h metabolite rhythms and so morning vs. evening studies are inherently limited in determining time of day variation with results typically limited to t-test/ANOVA analysis to determine significant changes between time points. These ‘gradient’ metabolites may or may not be rhythmic and a higher resolution (*n* > 4 samples per 24 h cycle) is required to discern rhythmicity.

Sato et al. [63] investigated the impact of a high-carbohydrate diet (HCD) and high-fat diet (HFD) of equal calorific content to observe the impact of nutritional challenge on the serum and skeletal muscle metabolome, whilst noting diurnal changes. Many time of day changes were observed as a result of the diet in the 1063 detected metabolite features (Table 2, Appendix A). For the HFD and HCD diet conditions 85/50 metabolites displayed a gradient change at baseline only, 126/95 post-completion of diet conditions only, and 138/142 both before and after diet conditions, respectively. Metabolites relating to lipid metabolism were enriched in the HFD group regardless of time of day—perhaps unsurprisingly. Conversely, half of all decreased metabolites in the evening as a result of HFD were related to amino acid metabolism despite equal calorific intake from protein between groups. Relative to HFD, HCD led to a decrease in metabolites relating to lipid metabolism, regardless of time of day. The authors suggested that increased serum insulin concentrations lead to suppressed lipolysis, explaining the observation, though the literature quoted in support of this statement observed this effect during high-activity (exercise) conditions. Sato et al. have demonstrated that diet composition, with regard to calorific intake from major food groups, displays an interaction with time of day variation of the metabolome altering specific metabolite rhythms; in the context of this study lipid metabolism was most strongly impacted.

#### 2.1.7. Morning vs. Evening Studies

Kim et al. [58] observed diurnal variation over 14 h in plasma (11 metabolites, 9%). The proportion of variance attributable to time of day in plasma was minimal vs. that attributed to patient ‘effects’ (age, sex, race, polycystic kidney disease ~40%) and residual variance (>50%). Data on sleep patterns, chronotype, work and light/dark history were not considered and so a proportion of this residual variance may well have a biological source, e.g., associated with chronotype, when considering the confidence in consistency and rigour by Kim et al. of their employed methodology.

Skarke et al. [60] observed diurnal variation in plasma (9/166, 5.4%) between 12 h samples over 48 h. Notably, less rhythmicity was observed when compared to the studies discussed above and may potentially be the result of reduced sampling frequency (12 hourly), the small cohort, or the purposeful monitoring of participants in their habitual routine undergoing unique and varied daily cycles. Despite the introduced inter-individual variation due to this setting, diurnal variation is still evident, suggestive of a high potency in temporal regulation of specific metabolites driven by external and internal rhythms. The reduced number of rhythmic metabolites from both of these studies compared to those discussed earlier aptly illustrates the difficulties in monitoring time of day variation and the required rigour in experimental design. To see beyond the noise and identify significant changes as a factor of time (clocks and/or circadian time), it is of importance to control environmental conditions, patient demographics and lifestyle.

### 2.2. Urine

Urine is the second most investigated biofluid, after blood, to assess time of day variation of the metabolome with six studies discussed here with the earliest work performed by Jerjes et al. [83], Walsh et al. [84] and Slupsky et al. [85], followed by several others [86,87] (see Table 3 and Appendix A).

#### 2.2.1. Sleep Deprivation and Prolonged Wakefulness

The participants in the Davies et al. study [51] also provided sequential urine samples across a 48 h study period [86]. Rhythmic metabolites (5/32, 15.6%) during the baseline sleep/wake cycle were observed, and 7/32 (22%), inclusive of the previous five, were observed during 24 h prolonged wakefulness. Eight metabolites significantly increased and a further eight decreased during sleep deprivation (Appendix A) with a relative concentration change ranging from −22.4% to +45.6%, the latter result comparable to Davies et al. [51]. Of the seven rhythmic metabolites identified during prolonged wakefulness, four remained significantly different to baseline around habitual wake time (07:00–09:00 h) but did not persist thereafter. The authors concluded that time of day is a more potent influencer on the urine metabolome than sleep deprivation; a similar conclusion given by Jerjes et al. [83] that sleep disturbances did not alter urinary steroid metabolite rhythms on the next day.

#### 2.2.2. Shift Work

Papantoniou et al. [87] set out to observe diurnal changes as well as define any significant differences between night and day shift workers in “sex hormones” for both male and female workers potentially brought about by circadian disruption/misalignment as a result of shift work (Appendix A). The potential consequences are associated with increased risk of developing breast and prostate cancer. Analysis of the full cohort revealed several progestagens and androgens which were significantly elevated in night shift workers vs. day shift workers, most notably observed within the subpopulation of pre-menopausal women. In day vs. night shift worker comparisons, testosterone, 3a,5a-androstanediol, 16-androstenol and pregnanediol were significantly elevated in pre-menopausal women and epitestosterone was elevated in post-menopausal women. There were no statistically significant differences in males. Peak time of androgens (epitestosterone, DHEA, etiocholanolone and 6a-hydroxyandrostenedione) were significantly later in the day in the night shift workers vs. day shift workers with the effect more pronounced in males vs. females.

#### 2.2.3. Creatinine

Walsh et al. [84] observed diurnal variation in the urinary metabolome, with creatinine being the prominent gradient metabolite and selectable marker within their constructed PLS-DA (partial least squares-discriminant analysis) model to predict time of day for collected samples, further commenting on the standardised diet reducing inter-individual variability. These findings were corroborated by Slupsky et al. [85] who also reported on a further five metabolites exhibiting diurnal variation and further supported by Kim et al. [58] (as described above, see Section 2.1.7) concluding that urine is susceptible to temporal and meal driven changes to the metabolome, more so than plasma. This increased temporal sensitivity may derive from circadian rhythms driven by the peripheral clock of the kidney resulting in time of day variation in renal function inclusive of diuresis [88], the method of sample collection (with Slupsky et al. collecting only two urine samples first void, and a second at 17:00 h) or the method of data normalisation to account for volume/concentration differences in provided urine samples.

A concern raised by Walsh et al., and Slupsky et al., is that urinary metabolite abundance is routinely normalised against creatinine, which shows inter-individual and diurnal variation, driven by diet/food consumption. Furthermore, it requires the assumption that the kinetics of excretion for metabolites of interest, which may change throughout the day, match that of creatinine thus resulting measurements of this normalisation may be less accurate than initially thought, as stated by Jerjes et al. [83]. Giskeødegård et al. [86], like Walsh et al., and Slupsky et al., observed diurnal variation of creatinine whilst additionally observing sleep deprivation to further impact creatinine levels thus undermining the role of creatinine for the purpose of normalisation as previously described. These concerns are supported by the findings of Jerjes et al. who concluded that creatinine did not undergo a significant daily rhythm when analysed independently via cosinor analysis, but androgen and cortisol metabolites did exhibit a significant daily rhythm thus by extension the relative ratio of cortisol/androgen metabolites:creatinine also exhibits a daily rhythm. This led Jerjes et al., to conclude that monitoring steroid/steroid or steroid/creatinine ratios is uninformative unless collection periods are timed as performed by Papantoniou et al. [87] and Giskeødegård et al. [86]. Other methods for normalisation of urine dilution are available [89].

### 2.3. Saliva

Time of day variation in saliva has garnered some coverage within the literature, perhaps due to ease of accessibility, with multiple laboratories reporting time of day variation (see Table 4) and various identified metabolites (Appendix A) across four independent studies.

#### 2.3.1. Circadian Variation

Dallmann et al. [56] (as described above, see Section 2.1.1) observed similar results in saliva as with plasma with ~15% (29 of 178) of the salivary metabolites displaying rhythmic variation, primarily consisting of amino acids and associated metabolites (Appendix A). Amino acids displayed a wide range of variation in abundance, up to ~400%, across 24 h. Their study therefore provides strong evidence that the salivary metabolome is influenced by circadian control, similar to their results with blood. Circadian variation should be a consideration in study design when analysing saliva due to the large magnitude of variation across the 24 h day. Despite the similar outcomes for blood/saliva samples, no identified rhythmic metabolites were common between these sample types, despite amino acids being a common rhythmic metabolite class observed in both sample types. It should be noted that sleep deprivation/prolonged wakefulness, which takes place during constant routine protocols, are associated with unique changes in the plasma metabolome and therefore may have a similar impact on the saliva metabolome but this has not yet been demonstrated.

#### 2.3.2. Morning vs. Evening Studies

Walsh et al. [84] (as described above, see Section 2.2.2), concluded that saliva exhibited diurnal variation with acetate being the prominent gradient metabolite and selectable marker within their constructed PLS-DA model to predict time of day for collected samples; the presence of which is attributed to acetate accumulation throughout the day due to carbohydrate fermentation in the mouth. No further gradient metabolites were reported in this study. Dallmann et al., observed extensive 24 h circadian rhythmicity compared to Walsh et al., who observed limited diurnal variation. This could be a result of Walsh et al. using ^1^H NMR and only analysing two samples (morning vs. evening), presumably ~12 h apart, compared to the MS methods and constant routine methodology (10 samples across 40 h) applied by Dallmann et al., Furthermore, Dallmann et al., employed various pre-study parameters to control for environmental/behavioural cycles and to reduce inter-individual variation which Walsh et al. reported was extensive within their study. Whilst the data were not published within the paper, Dame et al. [90] observed diurnal variation of acetate and amino acids across saliva samples collected in the morning and afternoon, corroborating with the findings of Walsh et al., and Dallmann et al., Skarke et al. [60] (as described above; see Section 2.1.7) reported 5.6% (14/250) of salivary metabolites display diurnal variation, notably less than what Dallmann et al. [56] observed under constant routine conditions, with suggested reasons for this disparity as previously described. It is therefore likely that circadian-controlled metabolite rhythms are masked in the diurnal setting of the Skarke et al., study; nevertheless, Skarke et al., shows time of day variation persists and is pronounced in a ‘real-world’ setting with gradient changes in metabolites still observable.

### 2.4. Breath

Of the five sample types discussed in this review, breath is the sparsest with regard to data on rhythmic/gradient metabolites. Only three studies have investigated time of day variation of breath (see Table 5 and Appendix A).

#### 2.4.1. Morning vs. Evening Studies

Sinues et al. [91] observed diurnal changes in breath across four time points (08:00 h–18:00 h) analysed via SESI-MS (secondary electrospray ionisation mass spectrometry) over nine consecutive days. The distinction between time points was great enough for machine learning approaches, in this case k-nearest neighbor validated via k-fold cross validation, to correctly predict sample time points 84% of the time in a blind classification. Unfortunately, the number of detected features, those which display gradient changes/exhibit diurnal variation, and metabolite identities were not elucidated.

#### 2.4.2. 24 h Diurnal Rhythms

A follow-up study saw Sinues et al. [92] observe diurnal variation in breath via a controlled laboratory study and using SESI-MS, where a total 111 features were analysed, of which 36–49% (average 40.3%) exhibited rhythmic behaviour. Pairwise comparisons differed drastically with regard to common features, a fact made more obvious due to the restricted size of the cohort (pilot study). A further limitation of this study, addressed by the author, is lack of identification of the detected features similar to the prior study, though some tentative metabolite identifications were provided but with no definitive conclusions in terms of the most rhythmic metabolite classes within the analysis nor the nature of such rhythms. Nevertheless, PCA score plots illustrate time of day variation within the samples.

Wilkinson et al. [93] monitored diurnal rhythms in volatile organic compounds within a cohort comprised of healthy and asthmatic participants. The authors incorrectly describe the observed rhythms as circadian; however, participants were not subject to a constant routine protocol thus observed rhythms should be defined as diurnal. Semantics aside, a key strength of this paper, compared to the work of Sinues et al. [92], is the identification of analysed metabolites with two, three and five metabolites observed in healthy, asthmatic, and combined groupings, respectively (Appendix A). Wilkinson et al., similar to Isherwood et al. [61] and Gehrman et al. [62], provides evidence for unique diurnal rhythms associated with specific phenotypes, primarily between healthy vs. asthmatic individuals. Unlike Isherwood et al., Wilkinson et al., did not observe any significantly rhythmic metabolites common to both groups when analysed separately.

### 2.5. Skeletal Muscle

The most recent human tissue to be investigated for time of day variation of the metabolome is skeletal muscle, with two of these three initial studies focussing on the lipidome (Table 6 and Appendix A).

#### 2.5.1. Diet Composition

Sato et al. [63] (as described above, see Section 2.1.6) compared the impact of a nutritional challenge on the skeletal muscle metabolome and concluded time of day (163/625 features affected), and thus accompanying environmental/behavioural cycles, more strongly influence the metabolome than diet composition (19/625 features affected). However, diet still exerted a significant influence, with HFD dampening the gradient changes of 60% of metabolites, predominantly related to lipid metabolism, with a further 19% of metabolites acquiring an inverted ‘gradient’, i.e., 19 metabolites with a higher relative abundance in the afternoon compared to the morning, or vice versa, displayed the opposite change post-HFD. Alternatively, the HCD saw metabolites related to lipid metabolism decrease in the morning and increase in the evening, creating a sharper gradient in relative abundance with a further 22% of metabolites exhibiting a significant difference in morning vs. evening samples (Appendix A).

#### 2.5.2. 24 h Diurnal Rhythms

Held et al. [95] performed a semi-targeted lipidomics assay showing that 13% of detected lipids (126/971) displayed significant rhythmicity over the 24 h day, comprising 57 (45%) glycerophospholipids, 52 (41%) diglycerides, 10 (8%) triglycerides, six (5%) sphingolipids and one (1%) sterol lipid(s). Loizides-Mangold et al. [94] also observed a high degree of rhythmicity amongst glycerophospholipids, sphingolipids and triacylglycerides though not diglycerides. An average of ~114 rhythmic metabolites (20.3% of detected lipids per participant) were observed by Loizides-Mangold et al., comparable to Held et al., with a reported 532 metabolites detected in all participants at all five time points and deemed comparable to an in vitro study run in parallel that analysed human myotube cultures. Of the rhythmic metabolites from the in vivo study it was reported that lipid levels altered by >20% across the 24 h time course with the authors drawing comparisons to similar findings in blood, saliva and urine reported elsewhere [51,53,56,86]. Of the rhythmic diglycerides and triglycerides analysed by Held et al., 87% and 60%, respectively, peaked at 04:00 h whilst 43% of the rhythmic sphingolipids peaked at 13:00 h, with similar observations made by Chua et al. [53] when studying plasma taken under constant routine conditions (see Section 2.1.2). Loizides-Mangold et al., however, reported sphingolipids peaking earlier at 04:00 h alongside phosphatidylcholines.

Further observations by Held et al. highlighted changes in rhythmicity of glycerophospholipids and fatty acids based on chain length (<20 and >20 carbon number) and degree of saturation, resulting in antiphase rhythms of fatty acids based on these parameters. Fatty acid chain length and level of saturation was only associated with larger amplitudes of observed rhythms with diglycerides, whilst sphingolipid and sterol lipid rhythms were deemed independent of these factors. These observations were only partially corroborated by Loizides-Mangold et al. who concluded the degree of saturation did not influence lipid rhythmicity but did state that chain length influences the diurnal profile of phosphatidylcholines and sphingomyelins.

The above studies clearly demonstrate that time of day variation is observed in the human muscle metabolome. Furthermore, not all lipid groups are affected equally and lipid chemical structure, in part, may impact how a given metabolite is regulated and influence the phase of its daily rhythm. The potential enrichment of specific metabolite groups at a given time of day should be a consideration both in study design as well as in data analysis and interpretation as temporal partitioning of specific subclasses may be misconstrued as relevant to the studied system when it is potentially an artefact resulting from time of sampling.

## 3. Discussion

Reviewing the literature has revealed the following key findings.

### 3.1. Key Findings

The number of studies investigating time of day variation of the human metabolome, to date, is small (*n* = 29).Endogenous metabolite rhythms, regulated by the circadian timing system, have been observed via constant routine studies in blood and saliva.Diurnal 24 h metabolite rhythms potentially evoked by external cues, either environmental (e.g., light/dark cycle) or behavioural (e.g., sleep/wake; feeding/fasting), have been observed in blood, urine, saliva, breath, and skeletal muscle.Acute changes in external cues, e.g., sleep/wake, feeding/fasting, activity/rest cycles and shift work, result in acute alterations to metabolite rhythms (timing and amplitude) that can persist after cessation of the change.Metabolite rhythms (timing and amplitude) may be sex dependent although sex has not been regularly investigated with regard to differences in 24 h metabolite rhythms.Specific physiological phenotypes and healthy vs. diseased state are shown to result in unique diurnal rhythms alongside the expected metabolite profiles of each phenotype.Lipids, in particular glycerophospholipids, and amino acids are the most frequently observed rhythmic metabolite classes. Lipid rhythms have shown the most variation between individuals with differences in phase (timing).Lipid rhythms may feature class-dependent temporal separation based upon carbon chain length and degree of saturation.A subset of metabolites are repeatedly reported as undergoing significant time of day variation across studies. A total of 35 putatively identified metabolites having been observed in at least five studies (Table 7) out of a total of 400 putatively identified across all studies.

### 3.2. Potential Consequences Resulting from Time of Day Variation

The above findings present a number of factors to consider in future metabolomics studies. Firstly, an individual participant can produce unique metabolic profiles with numerous significant differences between individual metabolites, even under constant conditions, from samples collected on the same day and only hours apart [53,56]. Time of day variation can confound intra- and inter-individual variation and may be significant enough to influence biological conclusions and biomarker identification. This example could also occur between two studies should samples have been collected at different times of day in turn affecting inter-study comparisons. Sampling participants at the same social/clock time is insufficient to circumvent this issue, e.g., as seen within the Chua et al. study [53] and discussed above in the introduction [44,45,46], individual participants can be sampled at the same social/clock time but still display inter-individual variation in their observed biological times due to genetic differences or differences in SCN and peripheral clock timing. Another topical example comes from recent studies proposing metabolite COVID-19 biomarkers [96]. While it remains to be shown whether daily variation has a significant impact on the proposed prognostic biomarkers, it should be noted that 28/77 metabolites identified as COVID-19 related are also reported in this review, and six of these have been observed undergoing time of day variation in at least five studies reported here (i.e., glutamic acid, isoleucine, kynurenine, leucine, ornithine, phenylalanine). With these issues in mind one should question to what extent biological rhythms are responsible for the observed results in studies applying metabolomics platforms such as a biomarker panel. That is not to say these studies are void for not considering biological time, however. Consider well-characterised rhythmic metabolites such as serotonin, tryptophan or melatonin and their use as biomarkers [97]. Serotonin and tryptophan concentration ranges are 0.04–0.74 µM and 35.60–121.67 µM (mean ± SEM 0.19 ± 0.01/72.24 ± 2.07), respectively, during a 24 h period (inclusive of 8 h sleep) [51]. Thus, controlling for biological/circadian time provides a means to disentangle variation brought about either by time of day or biological class (age, sex, disease state) or observe co-variance, as shown in this example [98]. Studies have shown that the timing of internal body clocks may differ between individuals by up to 12 h in urban areas of industrialised countries where shift work is common [99]. Therefore, scientific discoveries require further validation, in the context of biological time and variation across the 24 h day, before they can truly be relied upon. The work presented here is certainly not defining studies which do not include controls for diurnal and/or circadian rhythms as invalid, rather that when biomarkers are validated the time of day (clock time) and biological time (circadian time) of sample collection should be considered. The question to be asked is ‘Is the biomarker performance independent of the time of day and biological time the sample is collected’ or should specific requests for when a sample should be collected be applied? Providing context for biological time and acknowledgment that circadian timing systems impact on biological processes has started within other fields such as chronopharmacology [100]. It should be noted that the impact of time of day variation would be dependent upon the study design and method in which samples were collected, for example currently employed methods such as pooled 24 h urine samples circumvent time of day variation whilst blood samples collected in the morning following an overnight fast would reduce time of day variation, mitigate postprandial changes and minimise inter-individual variation with regard to biological time of the participants (assuming sampling occurred at a similar time, relative to waking up, for each participant/similar chronotypes between participants). These practices may not always be employed, however, an issue previously considered [101]. If sample collection occurs randomly throughout the day then the time of day variation in metabolites would lead to increased variation observed across all classes; larger cohort studies may exhibit a similar distribution of chronotypes and sampling time between classes translating to a comparable degree of variance between them with the caveat being that the investigated classes are independent, i.e., do not influence, chronotype distribution/sampling time. In such a scenario, the influence of time of day variation would be less, relative to a smaller cohort where assumptions on distributions could not be made, but there would still be a reduction in the sensitivity of any resulting statistical analysis to discern true significant differences between groups due to the introduced variation between/within groups as a result of time of day and potential covariance between the variable of interest and time. Therefore, smaller (pilot/discovery) studies not accounting for time of day variation are likely to be more strongly affected by the introduced variation. The metabolomics community could advance its research in a similar format through small additions and considerations during the study design process and updates to minimum reporting guidelines during publication. We hope with this review that time of day variation (driven by external factors and/or internal circadian timing) is given serious consideration in the future design of metabolomics and biomarkers studies so this effect is minimised or accounted for, thus, strengthening the design and interpretation of these studies.

### 3.3. Proposed Updates to Minimum Reporting Guidelines in Human Metabolomics Studies

Minimum reporting criteria were proposed by the Metabolomics Standards Initiative (MSI) [102] for various metabolomics studies and data analyses [103,104,105]. Whilst it was recognised that diurnal rhythms can influence the metabolome [103], this evidence was derived from animal studies with no pre-established protocols for human studies from which to derive a standardised workflow and guidelines. An underlying principle of the MSI guidelines is that all metadata that can reasonably be provided and that informs the metabolomics dataset must be made available [105]. Since the metabolome is influenced by circadian and diurnal rhythms, it is reasonable to, as a minimum, collect time data pertinent to these rhythms for the purpose of transparency and independent reproducibility, the purpose that these minimum standards were initially proposed for. This requirement for additional time information should also extend to in vitro studies with mammalian cell cultures responding to entraining agents [106,107] to express monitorable rhythms similar to in vivo studies [94] and prokaryotes also demonstrating 24 h rhythms [108,109] but is not discussed further in this paper. For studies in which single samples are being collected from participants self-reported questionnaires, such as the Munich Chronotype Questionnaire, should be administered to collect data on habitual sleep and assign participants their chronotype, in addition to recording work history specifically of those who are working non-traditional shifts outside 09:00–17:00 h work patterns (may differ with cultural context) or rotating shifts. These data may then inform cohort screening, e.g., exclusion of shift workers, or be retained simply for future reference. Most importantly, upon collecting a sample, elapsed time since participant wake up, complete calendar data, time of day, and approximate geographical location/coordinates at which samples were collected should be recorded. This latter measurement is important to assess the prevailing photoperiod (sunrise/sunset times) and is linked to an individual’s chronotype, their phase angle of entrainment, being more strongly linked to the sun clock (local time based on relative position of the sun) than the social clock (locally assigned time (time zones)). Context on the natural light/dark cycle that participants are entrained to therefore changes based on season and/or geographic location [99].

It has been suggested throughout the studies presented that observed diurnal rhythms (timing and amplitude) may differ based upon: age, sex and body mass, further emphasising the need for age/BMI/sex matched participants between test groups as is already common practice for many studies. Moreover, based on the findings of Honma et al. [50], diurnal rhythms and their response to an intervention (e.g., total sleep deprivation) can differ greatly between males and females and so facilitation should be made in the planning stages of data analysis to allow for sex-dependent comparisons within/between test groups to gauge such differences.

If opting to collect multiple samples over a time course then additional methodology, beyond that noted above, should be adopted from chronobiology studies such as monitoring and recording rest/activity and sleep/wake patterns, ideally for a week prior to sampling collected via actigraphy/sleep diary or monitoring a circadian-phase marker such as melatonin (plasma or saliva) or its derivative metabolite, aMT6s (6-sulphatoxymelatonin); methods of measuring such markers are reviewed here [110]. Having a circadian-phase marker, such as melatonin, allows for metabolite data collected over the time course to be plotted against biological time (e.g., against melatonin onset) as opposed to social time as demonstrated here [98,111]. Use of cosinor analysis and MetaCycle to determine rhythmicity is the standard approach within the field [112,113], though defining the amplitude of such rhythms is of equal importance. The amplitude of a 24 h metabolite rhythm and the minimum and maximum values represents the range of values a metabolite is present at in a sample. These data are therefore of great value and the publishing of such data for groups researching metabolite rhythms is encouraged with Davies et al. [51] setting an excellent example within their supplementary material by including minimum and maximum metabolite concentration values across their 48 h time course study which employed a targeted metabolomics assay.

### 3.4. Investigating Metabolite Rhythms—The Next Steps

A common outcome for many of the studies reviewed above was the identification of rhythmic/gradient metabolites; with a second common feature for many studies being the analytical platform employed. Upon observing the methods of the reviewed studies it becomes quickly apparent that particular assays have been collectively favoured with variations of LC-MS the most commonly used across 19 studies (two of which applied HILIC assay and 17 applied reversed- and/or normal-phase assays) followed by NMR (six studies), GC-MS (gas chromatography mass spectrometry) (six studies), FIA-MS (Flow injection analysis mass spectrometry) (four studies), SESI-MS (two studies), DI-MS (direct infusion mass spectrometry) (one study). Each of these platforms possesses its own advantages and limitations with regard to what metabolites can be detected thus influencing the collected dataset and metabolome coverage, resulting analysis and interpretation [1]. With this in mind, the curated list of metabolites undergoing time of day variation is somewhat limited and indeed biased towards lipids and non-polar metabolites which see better retention within reversed-phase assays. By contrast polar metabolites may be underrepresented since they exhibit poor retention in these assays and are better retained and detected within HILIC assays. Further utilisation of HILIC methodology may yield additional rhythmic metabolites similar to the outcomes of the study by Grant et al. [65]. It is also of interest to note that whilst various studies used multiple analytical platforms and assays in tandem, no study has yet to incorporate both HILIC and reversed-phase assays in parallel producing a more ‘complete’ dataset and coverage of the metabolome. Further untargeted studies employing a greater diversity of U(H)PLC (ultra high performance liquid chromatography) assays would benefit this growing body of work moving forward. Despite this potential limitation over 400 metabolites have been observed to be either significantly rhythmic or undergo significant time of day changes in a morning vs. evening fashion (Appendix A) based upon putative or definitive metabolite identification. Of these metabolites, 35 have been observed in at least five studies (Table 7). This compiled information provides insight into metabolic pathways likely influenced by time of day variation with amino acids and their derivatives being amongst the most frequently observed rhythmic/gradient metabolites. This information provides an adequate starting point for the development of bespoke targeted assays to investigate such pathways, quantify observed rhythms in metabolite concentrations similar to Davies et al. [51] and consider the biological significance of the rhythm, or lack thereof, under particular conditions such as shift work or disease state [61,64,67]. Expanding upon the number of biofluids/tissues investigated to observe diurnal/circadian variation presents a challenge due to the need for repeated and regular sampling over a 24 h time course, methods used for single sample collection, such as a biopsy, may thus prove too invasive or impractical for repeated and regular sampling. However, ambulatory microdialysis sampling techniques capable of high resolution sampling in humans hold promise [114]. In the near future it may be more practical then to expand upon the base of work on the five currently investigated sample types of which serum/plasma have been favoured leaving saliva, urine, skeletal muscle, and breath under investigated by relative comparison.

A general challenge for identifying and validating biological rhythms in -omic datasets across studies is the distinction between rhythmic and non-rhythmic time series. Many detection algorithms or combination of algorithms have been proposed. Two of these have already been mentioned above (i.e., cosinor and MetaCycle) but many other workflows and applications such as Multi-Omics Selection with Amplitude Independent Criteria (MOSAIC) [115], Rhythmicity Analysis Incorporating Non-parametric methods (RAIN) [116], Extended Circadian Harmonic Oscillator (ECHO) [117] and others [118,119] are available. As any one method is open to critique, MetaCycle, for example, is already combining a number of different methods to determine different regulation of rhythmicity in different groups of a study or between studies. Venn diagram analysis (VDA) employs any one of the methods above, e.g., RAIN, to identify changes in rhythmic items (transcripts/metabolites) between test groups. Recent findings, however, highlight inter-group and even inter-study comparisons where VDA has overestimated differences in rhythmic items [120]. The authors highlight the shortcomings of VDA and propose a novel approach to circumvent these issues implemented in the R package *compareRhythms* that compares circadian parameters (amplitude and phase) between the groups under comparison. This allows *compareRhythms* to discern between metabolites that have remained the ”same” (rhythmic across test conditions) and have undergone a ”change” (still rhythmic but phase/amplitude change between test conditions) whilst VDA cannot.

### 3.5. Summary

In summary, the primary objective of this review was to establish the sample types in which time of day variation of metabolite concentrations have been reported using a metabolomics platform, with a focus on identifying rhythmic metabolites. The extent of this time of day variation on the complete metabolome has also been reported to highlight the number of detected metabolites which have been shown to vary with time. The metabolome of blood, urine, saliva, breath, and skeletal muscle are influenced by diurnal and/or circadian rhythms. This most likely extends to other human biofluids and tissues in a similar fashion to how gene transcripts are rhythmic across a range of tissues in mammals [34,121]. Changes to external time cues (Zeitgeber), such as the light/dark and feeding/fasting cycle, result in changes to these rhythms and should thus be considered potential controllable variables, e.g., enforcing a specified light/dark, sleep/wake protocol for all participants, or setting meal times, excluding shift workers, dependent on the nature of the study being performed. Moving forward, additional data are suggested to be collected and shared within the metadata of metabolomics studies pertaining to history of shift work in participants, sleep/wake times and a person’s chronotype, complete time/calendar date and geographical location in which samples were taken, all of which may influence the metabolite profiles, the resulting analysis and biological interpretation.

## Figures and Tables

**Figure 1 metabolites-11-00328-f001:**
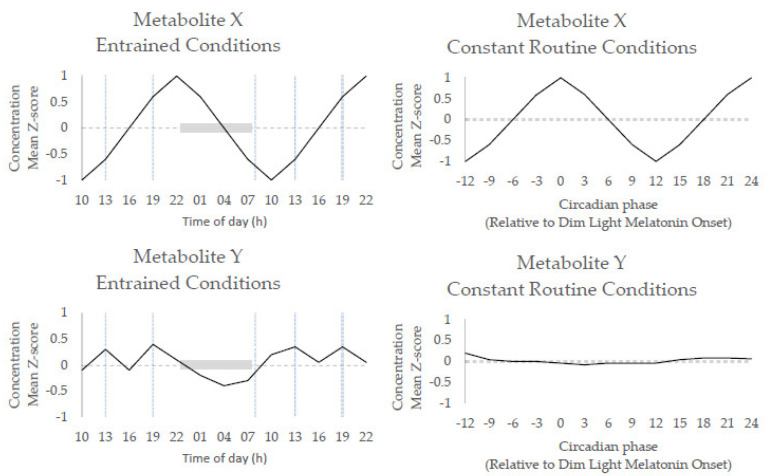
Mock data representation of two biological rhythms (e.g., metabolite rhythms) X (top) and Y (bottom) under entrained conditions (left) and constant routine (right). Entrained conditions consist of a light/dark cycle, meals at 08:00, 13:00, 19:00 h (shown by dashed vertical lines) and designated sleep time between 23:00 and 07:00 h (shown by grey shading of x-axis). (**Top-left**): Metabolite X: A rhythm with a regular 24 h period under entrained conditions may or may not be influenced by Zeitgeber (e.g., wake/sleep). (**Top-right**): Metabolite X: The rhythm observed under entrained conditions persists and maintains its 24 h periodicity under constant routine, the amplitude may or may not change between the conditions, and the rhythm is considered circadian. (**Bottom-left**): A rhythm with a more complex cycle, but regular 24 h period, and peaks correspond to mealtimes under entrained conditions (08:00, 13:00, 19:00 h), suggesting some effect of feeding/fasting cycles. (**Bottom-right**): The rhythm is significantly dampened in constant routine (does not persist), with a regular period/amplitude no longer detectable. The rhythm of metabolite Y is not considered circadian in nature, as it did not persist under constant routine conditions, and is classed as a diurnal rhythm, i.e., a rhythm evoked by exogenous cycles such as feeding/fasting and sleep/wake.

**Figure 2 metabolites-11-00328-f002:**
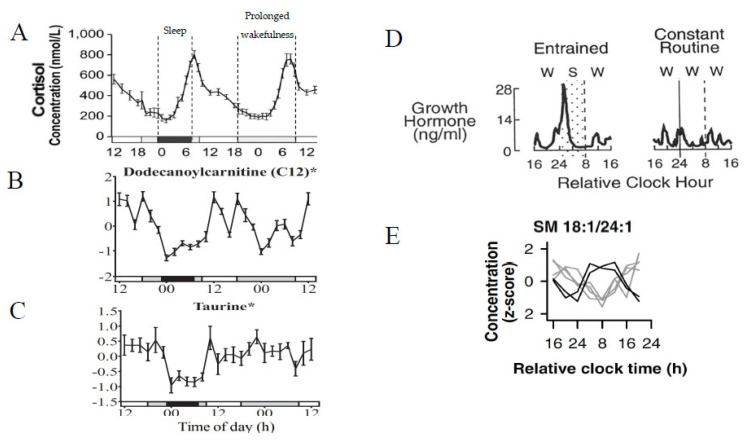
Comparative metabolite/hormone profiles under entrained conditions: sleep vs. prolonged wakefulness (**A**–**C**), entrained vs. circadian constant routine conditions (**D**), and inter-individual variation under constant routine conditions (**E**), reproduced from Honma et al. [50] (**A**), Davies et al. [51] (**B**,**C**), Czeisler & Klerman [52] (**D**), and Chua et al. [53] (**E**). A,B,C: Comparative profiles of cortisol, dodecanoylcarnitine (C12), and taurine under entrained light/dark conditions, sleep (highlighted in black) vs. prolonged wakefulness (highlighted in grey), with meals provided at 07:00 h, 13:00 h, 19:00 h, and 22:00 h (snack). No significant difference observed in cortisol, a SCN-driven hormone, between sleep conditions (**A**), peaks in measured intensity (y-axis) corresponding to mealtimes observable in various lipids and amino acids (**B**,**C**), alongside statistically significant perturbations during prolonged wakefulness vs. sleep [51]. **D**: Growth hormone rhythm observable under entrained conditions (peak during sleep) but dampened under constant routine. E: Individual rhythmic profiles of six participants showing inter-individual variation in lipid profiles (SM18:1/24:1), with two individuals displaying an inversed rhythm (~12 h ahead/delayed) relative to the other four participants.

**Table 1 metabolites-11-00328-t001:** Summary of literature search parameters and outcomes. Two separate searches were performed, with the first search featuring ‘circadian’ as a key word (performed circa 21 April 2020) and the second search featuring ‘diurnal’ in lieu of circadian (highlighted—light grey, performed circa 23 July 2020). The two searches were performed to encompass as much literature as possible observing daily rhythms. On the first search, the search terms had to be reduced when using Google Scholar due to insufficient ‘hits’ (*n* = 1 ‘hits’) resulting from excess search terms. Reduced hits on the second search when using the phrases ‘diurnal’/’diurnal variation’ may stem from neither term corresponding to MeSH terms (for PubMed) and potentially being uncommon key words associated with literature from other search engines reducing discoverability. Manual searches consisted of looking for prior referenced work in collected literature that met inclusion criteria.

Search Terms	Database/Search Engine	‘Hits’	Relevant Papers (Based on Abstract)	Met Inclusion Criteria *	
				Circadian Studies	Diurnal Studies
“Human(s)” “Circadian Rhythm OR Circadian Clocks” “Metabolomics OR Metabolome” **	PubMed (NCBI)	70	133	6	19
	Web of Science	52			
“Metabolomic” “Circadian” “Rhythm” “Human” “Chronobiology”	Google Scholar	212			
N/A	Further manual searches	13			
“Human(s)” “Diurnal Variation OR Diurnal”, “Metabolome OR Metabolomics” ***	PubMed (NCBI)	19	123 (Majority duplicates of prior search)	3	16
	Web of Science	28			
“Metabolomic” “Metabolome” “Diurnal” “Rhythm” “Human” “Chronobiology”	Google Scholar	92			
N/A	Further manual searches	0			

*A total of 29 novel papers met inclusion criteria, human studies employing a metabolomics platform taking multiple samples over a time course** Search terms corresponding to MeSH terms; *** No MeSH terms corresponding to “Diurnal” OR “Diurnal variation”.

**Table 2 metabolites-11-00328-t002:** A brief summary of study design, cohort details and results with regard to observed time of day variation of metabolites for relevant studies analysing plasma/serum.

Author(s)	Assay/Platform	Time Course Details	Study Setting/Conditions	Cohort Details	Rhythmic/Gradient Metabolites/ Features Observed	Rhythmic/Gradient Classes Primarily Observed
Park et al., (2009) [54]	Untargeted ^1^H NMR	Diurnal variation 24 h, 1 h intervals between samples	‘Inpatient’ Standardised meals. Consistent light/dark cycle	N = 10, 5 males Age 22–83 BMI 18.5–32.6	34	Amino acids Lipids (unidentified)
Ang et al., (2012) [55]	Untargeted UPLC/Q-TOF MS (Reversed Phase)	Diurnal variation (25 h, 3 h intervals between samples)	‘Inpatient’ 17:8 wake/sleep, light/dark cycle. Hourly isocaloric meals Semi-recumbent position	N = 8 All male Age 53.6 ± 6.0 BMI 23.2 ± 1.4	203 features (19%) 34 metabolites	Amino acids Acylcarnitines LysoPEs LysoPCs
Dallmann et al., (2012) [56]	Untargeted GC-MS LC-MS (Reversed Phase)	Circadian variation (constant routine 40 h, 4 h intervals between samples)	‘Inpatient’ Standard constant routine parameters (see [41])	N = 10 (split into 2 equal groups, within which samples were pooled for each 4 h interval) All male Age 57.8 ± 1.0 & 61.0 ± 0.6 BMI 26.6 ± 0.6 & 25.1 ± 0.5	41 (15%)	Amino acids Glycerophospholipids Acylcarnitines Steroid hormones
Kasukawa et al., (2012) [57]	Untargeted LC-TOF MS (Reversed Phase)	Circadian variation (forced desynchrony 28 h, bookended by constant routine protocols (38 h each, 2 h intervals between samples)	‘Inpatient’ Standard constant routine parameters (with the exception of meals every 2 h (see [41]) Controlled light/dark cycles, temperature during forced desynchrony	N = 6 All male Aged 20–23	312 features (7%)	Amino acids Steroid hormones
Chua et al., (2013) [53]	Targeted Lipidomics LC-MS/MS (Reversed Phase)	Circadian variation (constant routine 37 h, 4 h intervals between samples at 5 h onwards of constant routine)	‘Inpatient’ Standard constant routine parameters (see [41])	N = 20 All male Age 24.4 ± 1.8 3 ‘Overweight’ 17 ‘Healthy	35 (13.3%)	Glycerolipids Glycerophospholipids
Davies et al., (2014) [51]	Untargeted UPLC/Q-TOF MS/MS and targeted FIA-MS UPLC-MS/MS (Reversed Phase)	Diurnal variation (24 h). 24 h wake/sleep cycle vs. 24 h prolonged wakefulness, 2 h intervals between samples 48 h	‘Inpatient’ Standardised meals and mealtimes. Controlled light/dark cycle and activity/posture	N = 12 All male Age 23 ± 5, BMI 24.5 ± 2.3	109 (63.7%) sleep/wake 88 (51.5%) sleep deprivation 78 (45%) during both conditions	Amino acids Acylcarnitines LysoPCs Phosphatidylcholines Sphingolipids Fatty acids
Kim et al., (2014) [58]	Untargeted LC—TOF MS (Reversed Phase)	Diurnal variation Sampling 1, 3, 7, 9, 11, 14 h post-wake, first sample fasted.	‘Inpatient’ Standardised meals and mealtimes	N = 26 14 males Age 33 ±10.9 BMI 24.3 ±3.3	11 (9%)	LysoPCs Phosphatidylinositol
Chua et al., (2015) [59]	Targeted Lipidomics LC-MS/MS (Reversed Phase)	Circadian variation (constant routine 37 h, 4 h intervals between samples at 5 h onwards of constant routine)	‘Inpatient’ Standard constant routine parameters (see [41]}	N = 20 All male Age 23 ± 5 BMI 24.5 ± 2.3	4 (1.5%) decreased during sleep deprivation 21 (5.5%) increased during sleep deprivation	Sphingomyelins TAGs Phosphatidylcholines Phosphatidylinositol
Skarke et al., (2017) [60]	Targeted LC-MS/MS (HILIC)	Diurnal variation am vs. pm (48 h, 5 samples 12 h apart)	‘Outpatient’	N = 6 All male Age 32.3 ± 3.6 BMI 25.2 ± 3.4	9 (5.4%)	
Isherwood et al., (2017) [61]	Targeted FIA-MS UPLC-MS/MS (Reversed Phase)	Diurnal variation (24 h—2 h intervals between samples)	‘Inpatient’ Controlled sleep/wake, light/dark cycle, and posture Hourly isocaloric meals	N = 23 All male BMI/Age Lean group 23.2 ± 1.4/53.6 ± 6.0 OW/OB 29.8 ± 2.3/51.0 ± 7.7 T2DM group 31 ± 1.6/57.3 ± 4.8	50/130 (38.5%) total 35—lean 39—OW/OB 20—T2DM	Amino acids Phosphatidylcholines LysoPCs Acylcarnitines
Gehrman et al., (2018) [62]	Targeted ^1^H NMR	Diurnal variation (48 h—2 h intervals between samples)	‘Inpatient’ Habitual sleep/wake cycle Hourly isocaloric meals	N = 30 20 male and 10 females (split equally into 2 groups) BMI < 29 Healthy Age 35.0 ± 7.5 Insomnia Age 37 ± 7.9	24 (total) 11 common to both groups 6 unique to healthy 7 unique to insomnia	Amino acids
Sato et al., (2018) [63]	Untargeted UHPLC-MS/MS GC-MS	Diurnal variation am vs. pm	‘Outpatient’ Standardised meals and mealtimes	N = 8 All male Age 30–45 BMI 27–32.5	532, 130, 349 features (50%, 12%, 33%) time of day, diet, time of day diet interaction, respectively. After HFD 13% features lost daily variation, 17% gained new daily variation After HCD 7% features lost daily variation 14% gained new daily variation	Amino acids Fatty acyls Glycerolipids Glycerophospholipids Sphingolipids Carbohydrates Xenobiotics
Skene et al., (2018) [64]	Targeted FIA-MS UPLC-MS/MS (Reversed Phase)	Circadian variation (constant routine 24 h, 11 samples at 1–3 h intervals) Day shift vs. night shift (simulation)) circadian vs. behavioural control	‘Inpatient’ Standard constant routine parameters (see [41]) During baseline & shift work—controlled sleep/wake, light/dark cycle, temperature. Standardised meals and mealtimes	Night shift: N = 7 6 males Age 27.6 ± 3.2 BMI 25.6 ± 3.3 Day shift: N = 7, 4 males Age 24.0 ± 2.2 BMI 25.9 ±3.4	65 (49.2%) across both shift patterns, 27 (20.5%) common to both	Amino acids LysoPCs Phosphatdylcholines Acylcarnitines Glycerophospholipids Sphingolipids
Grant et al., (2019) [65]	Untargeted & Targeted LC-QTOF/MS (HILIC)	Circadian variation (24 h) Circadian- vs. wake-dependent changes	‘Inpatient’ Standard constant routine parameters (see [41])	N = 13 9 males Age 25.0 ± 4.3 BMI 22.0 ± 2.1	Targeted: Group level 28/99 (28.3%) (rhythmic, rhythmic & linear) 4/99 (4%) linear Untargeted: Group level 361 (22%) rhythmic features 8% linear features Individual level 14% rhythmic profiles 4% linear profile	Amino acids Organic acids
Gu et al., (2019) [66]	Untargeted UHPLC-MS (Reversed phase) & GC-MS/MS	Diurnal variation (26–48 h) (48 h time course for N = 2, 26 h for N = 1 participants),	‘Inpatient’ Standardised meals and mealtimes Habitual sleep time (10 h sleep)	N = 3 2 males Age 20–31 BMI 18 < 29.9	100/663 (15.1%) rhythmic in at least 1 individual 26/663 (3.9%) rhythmic in at least 2 individuals.	Amino acids DAGs Lysolipids Phospholipids Steroid lipids
Kervezee et al., (2019) [67]	Targeted DI-MS LC-MS/MS (Reversed phase)	Diurnal variation (24 h—2 h intervals between samples) Baseline vs. forced misalignment post-simulated shift work	‘Inpatient’ Controlled sleep/wake, light/dark cycle and hourly isocaloric meals during sampling periods	N = 9 8 males Age 22.6 ± 3.4 BMI 21.3 (19.6–23)	51 (39.2%) baseline 53 (40.8%) night shift 32 (24.6%) both, 24 phase shifted, 27 (21%) significantly changed post-night shift	Amino acids Fatty acids Organic acids Lysophospholipids PCs
Honma et al., (2020) [50]	Targeted FIA-MS UPLC-MS/MS (Reversed Phase)	Diurnal variation (70 h, 2 h intervals between samples) 16:8 wake/sleep cycle > 40 h prolonged wakefulness > 8 h recovery sleep	‘Inpatient’ Standardised meals and mealtimes. Controlled light/dark cycle and activity/posture	N = 12 All female Age 25 ± 4 BMI 24.9 ± 3.6	Total 97/130, 58 (44.6%) common for all conditions. Baseline 78 (60%) 8 unique. Sleep deprivation 76 (58.5%) 5 unique Recovery sleep 80 (61.5%) 5 unique	Glycerophospholipids Sphingolipids Amino acids Biogenic amines Acylcarnitines
Lusczek et al., (2020) [68]	Untargeted UHPLC/MS (Reversed Phase)	Diurnal variation (24 h—4 h intervals between samples)	‘Inpatient’ Self-selected light/dark, feeding/fasting, sleep/wake cycle for healthy participants	Healthy cohort N = 5 2 males, Age 45–72 BMI 22.4–33.3 ICU cohort N = 5 2 males Age 43–66 BMI 31.0–57.3	10 (16.7%) in healthy 0 in ICU	Amino acids Acyl carnitines LysoPEs

Footnotes: Age and BMI are quoted in standard units, years and kg/m^2^, respectively. Where available, mean age/BMI ± 1 SD given. Significant changes in metabolites identified in studies performing am vs. pm (two-time point) comparison(s), should be considered as gradient changes ergo ‘gradient metabolite’, Significant changes in metabolites identified in studies over a >24 h time course with *n* ≥ 5 should be considered as rhythmic changes ergo ‘rhythmic metabolite’, rhythmicity being detected by cosinor analysis and/or MetaCycle. Rhythmic/gradient features are denoted as such, otherwise the table refers to rhythmic/gradient metabolites. Rhythmic/gradient classes primarily observed are not an exhaustive list of all metabolite classes observed within a study but a summary of the most rhythmic classes, if any, for that particular study, as denoted by the author or inferred from provided data. Abbreviations: DAG—diglyceride; DI-MS—direct infusion mass spectrometry; FIA-MS—flow injection analysis mass spectrometry; GC-MS—gas chromatography mass spectrometry; HILIC—hydrophilic interaction chromatography; LC-MS—liquid chromatography mass spectrometry; LysoPC—lysophosphatidylcholine; LysoPE—lysophosphatidylethanolamine; MS/MS—tandem mass spectrometry; NMR—nuclear magnetic resonance; PC—phosphatidylcholine; Q-TOF MS—quadruple time of flight mass spectrometry; SESI-MS—secondary electrospray ionisation mass spectrometry; TOF MS—time of flight mass spectrometry; U(H)PLC—ultra high performance liquid chromatography.

**Table 3 metabolites-11-00328-t003:** A brief summary of study design, cohort details and results with regard to observed time of day variation of metabolites for relevant studies analysing urine.

Author(s)	Assay/Platform	Time Course Details	Study Setting/Conditions	Cohort Details	Rhythmic/Gradient Metabolites/Features Observed	Rhythmic/Gradient Classes Primarily Observed
Jerjes et al., (2006) [83]	Targeted GC-MS	Diurnal variation (24 h—3 h intervals between samples)		N = 20 10 males Age 32 ± 5.4 BMI 23.5 ± 2	9	Androgens Cortisol metabolites
Walsh et al., (2006) [84]	Untargeted ^1^H NMR	Diurnal variation am vs. pm	‘Outpatient’ Standardised meals	N = 60 30 males Age 19–69	1	
Slupsky et al., (2007) [85]	Targeted ^1^H NMR	Diurnal variation am vs. pm	‘Outpatient’	N = 30 23 females Age 24.7 ± 2.7 BMI 22.7 ± 0.97	6	
Kim et al., (2014) [58]	Untargeted LC—TOF MS (Reversed Phase)	Diurnal variation Sampling 1, 3, 7, 9, 11, 14 h post-wake, first sample fasted.	‘Inpatient’ Standardised meals and mealtimes	N = 26 14 males Age 33 ± 10.9 BMI 24.3 ± 3.3	135 (46%)	Glycerophospholipids LysoPCs Phosphatidylinositol
Giskeødegård et al., (2015) [86]	Untargeted ^1^H NMR	Diurnal variation (48 h) Samples at 2–4 h intervals when awake, 8 h overnight	‘Inpatient’ Standardised meals and mealtimes. Controlled light/dark cycle and activity/posture	N = 15 All male Age 23.7 ± 5.4	5 (15.6%)—sleep/wake cycle 7 (22%) during 24 h wakefulness During sleep deprivation 8 increased, 8 decreased	Amino acids Fatty acids
Papantoniou et al., (2015) [87]	Targeted GC-MS	Diurnal variation (24 h)	‘Outpatient’ Day vs. night shift workers	N = 117 63 males Age 22–64 BMI 22.6–30.6	5 (31.3%) significantly different in premenopausal day vs. night workers	Progestagens Androgens

Footnotes: See Table 2. Abbreviations: GC-MS—gas chromatography mass spectrometry; LC-MS—liquid chromatography mass spectrometry; LysoPC—lysophosphatidylcholine; NMR—nuclear magnetic resonance; TOF MS—time of flight mass spectrometry.

**Table 4 metabolites-11-00328-t004:** A brief summary of study design, cohort details and results with regard to observed time of day variation of metabolites for relevant studies analysing saliva.

Authors	Assay/Platform	Time Course Details	Study Setting/Conditions	Cohort Details	Rhythmic/Gradient Metabolites/Features Observed	Rhythmic/Gradient Classes Primarily Observed
Walsh et al., (2006) [84]	Untargeted ^1^H NMR	Diurnal variation am vs. pm	‘Outpatient’ Standardised meals	N = 60 30 males Age 19–69	1	No gradient metabolite classes identified
Dallmann et al., (2012) [56]	Untargeted GC-MS LC-MS (Reversed Phase)	Circadian variation (constant routine 40 h, 4 h intervals between samples)	‘Inpatient’ Standard constant routine parameters (see [41])	N = 10 (split into 2 equal groups within which samples were pooled for each 4 h interval) All male Age 57.8 ± 1.0 & 61.0 ± 0.6 BMI 26.6 ± 0.6 & 25.1 ± 0.5	29 (15%)	Amino acids
Dame et al., (2015) [90]	Untargeted ^1^H NMR	Diurnal variation sampling at prebreakfast vs. 2 h post-breakfast vs. 2 h post-lunch		N = 16 8 males & females Age (24–42) (only N = 2 took part in observation of diurnal variation)	8 (10.5%)	Amino acids
Skarke et al., (2017) [60]	Targeted LC-MS/MS (HILIC)	Diurnal variation am vs. pm (48 h, 5 samples 12 h apart)	‘Outpatient’	N = 6 All male Age 32.3 ± 3.6 BMI 25.2 ± 3.4	14 (5.6%)	Amino acids

Footnotes: See Table 2. Abbreviations: GC-MS—gas chromatography mass spectrometry; HILIC—hydrophilic interaction chromatography; LC-MS—liquid chromatography mass spectrometry; MS/MS—tandem mass spectrometry; NMR—nuclear magnetic resonance.

**Table 5 metabolites-11-00328-t005:** A brief summary of study design, cohort details and results with regard to observed time of day variation of metabolites for relevant studies analysing breath.

Authors	Assay/Platform	Time Course Details	Study Setting/Conditions	Cohort Details	Rhythmic/Gradient Metabolites/Features Observed	Rhythmic/Gradient Classes Primarily Observed
Sinues et al., (2012) [91]	Untargeted SESI-MS	Diurnal variation (4 time periods) 8:00–11:00, 11:00–13:00, 13:00–15:00, 15:00–18:00	‘Outpatient’	N = 12 7 males	Diurnal changes observed but number of rhythmic features not reported	No metabolites structurally identified
Sinues et al., (2014) [92]	Untargeted SESI-MS	Diurnal variation (24 h, 1 h intervals, 5–7 repeats per sample)	‘Inpatient’ Controlled laboratory conditions: hourly isocaloric meals, constant wakefulness, consistent light conditions	N = 3 2 males Age 33–38	40 (36%) of features (49% in N = 1)	No metabolites structurally identified
Wilkinson et al., (2019) [93]	Untargeted GC-MS	Diurnal variation (24 h—4 time points: 16:00, 22:00, 04:00, 10:00)	Standardised meals and feeding schedule. Maintained habitual bedtime	Healthy N = 10 7 males Age 27.5–49.3 BMI 23.4–30.5 Asthma N = 9 7 male Age 26.0–49.5 BMI 22.3–27.2	Combined dataset 5/102 (4.9%) metabolites Asthma 3/102 (~2.9%) metabolites, 1 of which is unique to this group in addition to rhyth-micity of exhaled nitric oxide fraction Healthy 2/102 (~2%) metabolites rhythmic and unique to this group	Volatile organic compounds

Footnotes: See Table 2. Abbreviations: GC-MS—gas chromatography mass spectrometry; SESI-MS – secondary electrospray ionisation.

**Table 6 metabolites-11-00328-t006:** A brief summary of study design, cohort details and results with regard to observed time of day variation of metabolites for relevant studies analysing skeletal muscle.

Authors	Performed Assay	Time Course Details	Study Setting/Conditions	Cohort Details	Rhythmic/Gradient Metabolites/Features Observed	Rhythmic/Gradient Classes Primarily Observed
Loizides-Mangold et al., (2017) [94]	Targeted (Lipidomics) LC-MS	Diurnal variation (24 h—4 h intervals between samples)	‘Inpatient’ Controlled sleep/wake, light/dark cycle, temperature. Isocaloric meals	N = 10, 9 males Age 29.9 ± 9.8 BMI 24.1 ± 2.7	106 of 1058 metabolites (10%)	TAGs, PCs, Pes PIs, PSs, CLs Cers, GlcCers, SMs
Sato et al., (2018) [63]	Untargeted UHPLC-MS/MS GC-MS	Diurnal variation am vs. pm	‘Outpatient’ Standardised meals and mealtimes	N = 8, All male Age 30–45 BMI 27–32.5	163 & 19 of 625 features (26% & 3%) as a result of time of day & diet, respectively	Amino acids Fatty acyls Glycerolipids Glycerophospholipids Sphingolipids Carbohydrates Xenobiotics
Held et al., (2020) [95]	Semi-targeted Lipidomics UPLC/HRMS (reversed & normal phase)	Diurnal variation (24 h—5 h intervals between samples)	‘Inpatient’ Controlled sleep/wake, light/dark cycle. Standardised meals and mealtimes	N = 12, All male Age 22.2 ± 2.3 BMI 22.4 ± 2.0	126 of 971 (13%)	Glycerophospholipids TAGs Sphingolipids DAGs Sterol Lipids

Footnotes: See Table 2. Abbreviations: Cer—ceramide; CL—cardiolipin; DAG—diglyceride; GC-MS—gas chromatography mass spectrometry; GlcCer—glucosylceramide; HRMS—high resolution mass spectrometry; LC-MS—liquid chromatography mass spectrometry; MS/MS—tandem mass spectrometry; PC—phosphatidylcholine; Pe—phosphatidylethanolamine; PI—phosphatidylinositol; PS—phosphatidylserine;; SM—sphingomyelin; TAG—triglyceride; U(H)PLC—ultra high performance liquid chromatography.

**Table 7 metabolites-11-00328-t007:** Putatively identified metabolites, observed in five or more human metabolomics time course studies, that underwent significant time of day variation (rhythmic/gradient metabolites) in ranked order.

Rank	Putative Identification of Rhythmic/Gradient Metabolites	InChIKey	Number of Studies Significant Changes were Observed in
1	Proline	ONIBWKKTOPOVIA-BYPYZUCNSA-N	11
2	Leucine	ROHFNLRQFUQHCH-YFKPBYRVSA-N	10
3	PC(32:0)	-	10
4	Phenylalanine	COLNVLDHVKWLRT-QMMMGPOBSA-N	9
5	Ornithine		9
6	Tyrosine	OUYCCCASQSFEME-QMMMGPOBSA-N	9
7	Glutamic acid	WHUUTDBJXJRKMK-VKHMYHEASA-N	8
8	Isoleucine	AGPKZVBTJJNPAG-WHFBIAKZSA-N	8
9	LysoPC(18:2) and/or LysoPE (18:2)	-	8
10	PC(34:3)	-	8
11	Citrulline	RHGKLRLOHDJJDR-BYPYZUCNSA-N	7
12	Taurine	XOAAWQZATWQOTB-UHFFFAOYSA-N	7
13	Tryptophan	QIVBCDIJIAJPQS-VIFPVBQESA-N	7
14	Valine	KZSNJWFQEVHDMF-BYPYZUCNSA-N	7
15	LysoPC(18:1)	-	6
16	LysoPC(16:0)	-	6
17	Aminoadipic acid	OYIFNHCXNCRBQI-BYPYZUCNSA-N	6
18	Citric acid	KRKNYBCHXYNGOX-UHFFFAOYSA-N	6
19	Cortisone	MFYSYFVPBJMHGN-ZPOLXVRWSA-N	6
20	Creatinine	DDRJAANPRJIHGJ-UHFFFAOYSA-N	6
21	Glycine	DHMQDGOQFOQNFH-UHFFFAOYSA-N	6
22	Kynurenine	YGPSJZOEDVAXAB-UHFFFAOYSA-N	6
23	PC C36:2	-	6
24	Alanine	QNAYBMKLOCPYGJ-REOHCLBHSA-N	5
25	Cortisol	JYGXADMDTFJGBT-VWUMJDOOSA-N	5
26	Lysine	KDXKERNSBIXSRK-YFKPBYRVSA-N	5
27	LysoPC(17:0)	-	5
28	PC C34:1	-	5
29	PC C34:2	-	5
30	PC(32:1)	-	5
31	Pregnenolone sulfate	DIJBBUIOWGGQOP-OZIWPBGVSA-N	5
32	Sarcosine	FSYKKLYZXJSNPZ-UHFFFAOYSA-N	5
33	SM(20:2)	-	5
34	Threonine	AYFVYJQAPQTCCC-GBXIJSLDSA-N	5
35	Trimethylamine N-oxide (TMAO)	UYPYRKYUKCHHIB-UHFFFAOYSA-N	5

Footnotes: Common names have been assigned to act for synonyms reported in the literature; see Appendix A for further details. Putative identifications are ordered based on the frequency in which they are reported within the literature, with metabolites only listed here if observed in n ≥ 5 studies; InChlKeys are provided where applicable. Abbreviations: PC—phosphatidylcholine; LysoPC—lysophosphatidylcholine; LysoPE—lysophosphatidylethanolamine; SM—sphingomyelin.

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
