# Peer review of "Tick-Tock Consider the Clock: The Influence of Circadian and External Cycles on Time of Day Variation in the Human Metabolome—A Review"

_metabolites, 2021, doi:10.3390/metabo11050328_

Round 1

Reviewer 1 Report

The manuscript by Hancox et al. entitled: “Tick-Tock-Consider the Clock: The Influence of Circadian and External Cycles on Time of Day Variation in the Human Metabolome – A Systematic Review”, is a comprehensive review of the metabolomics studies investigating time of the day variation in metabolome. The topic is interest, as the temporal changes in metabolome may be a confounding factor in biomarker discovery (by generating additional variance), but at the same time when studied properly may carry novel insights into whole body regulatory processes. The article is divided into sections describing different biofluids and this structure is appropriate in my opinion as it allows authors to organize tables in biofluid specific way. It is clear that this article has been written by the experts in the field of chronobiology and metabolomics. The manuscript is well written and not only reports the data from the literature but provides additional narration to these findings and highlights future direction for human metabolomics studies. While, the very systematic approach of describing exact numbers (or %) of metabolites identified to be rhythmic in various publications becomes little tedious to follow at times, there is no question that this manuscript will become a great resource for future metabolomics studies in general, not only chronobiology focused. Authors, point towards, some important aspects of studying metabolome which needs to be addressed in the future, and propose minimum reporting guidelines to account for time of the day variation. I have few comments which should be addressed prior to considering this manuscript for publication.

Major comments:

  1. Although the article is focused on human circadian metabolic rhythms, greater acknowledgement of animal studies in introduction and discussion would be beneficial. Substantial amount of rodent research on clock genes exists, and it could be used as a strong mechanistic support of the ideas presented in this paper.
  2. One aspect which seems to be overlooked by authors is the fact that for many metabolomics studies samples are typically collected in the morning after the overnight fast (similar to standard blood biochemistry sample collection). While, this does not apply to all studies or remove effect of rhythmic metabolites, it may substantially lower the variance in many published reports comparing to the situation where samples would be collected randomly throughout the day in either postprandial or fasting state. I suggest adding few sentences to discuss this inherent characteristic of many metabolomics studies.
  3. Consider adding a figure with either the definitions or visualization of the most important terms such as circadian and diurnal rhythm. Some form of graphical representation of these patterns could be useful for the readers to avoid the confusion and mistakes, which apparently are common in the literature (Page 3, line 125-126).
  4. Maybe I missed this, but could authors provide the definition of “gradient metabolite”?
  5. Many studies are referenced in the text using only first author name, but probably different format should be used: “Author Last Name et al.“ For example: “Grant et al. [54]”.

Minor comments:

  1. Page 2, line 62, This is the first table cited in the text, but it has number 7, please make sure to reference tables in order of appearance in the text.
  2. Page 2, lines 92-94. Not all readers may be familiar with Q10 temperature coefficient. Please consider rewriting this part to better explain Q10 in relationship to period of circadian rhythm.
  3. Page 3, line 100, please provide reference for the “gold standard protocol”.
  4. Page 7, line 291, use superscript for “1” in 1H NMR.
  5. Page 8, line 309, Phrase “healthy vs unhealthy” is very general, consider using “…physiological and pathological states caused by disease can lead to distinct metabolic profiles.”.
  6. Page 9, line 363, please consider using just “high carbohydrate diet” instead of “high carbohydrate content diet”. Similarly change to high fat diet (HFD).
  7. Page 9, line 372, “HCD lead to decrease in …” comparing to what? this sentence is unclear.
  8. Figures S1, and S2. Please consider providing short study references in addition to citations in the figure caption. For example, in Figure S2, instead of “E [54]” consider using Grant et al. [54]” This would be helpful to the reader, as Authors refer to some of these studies multiple times in the text.

Author Response

Please see the attachment for author responses. Responses specific to "Reviewer 1" are on pages ~1-4.

Reviewer 2 Report

This is an interesting and thorough review of all studies to date on the influence of circadian and external cycles on the time of day variation in human metabolites.  In my opinion the manuscript is ready for publication.  I only have a few comments/corrections to suggest:

- General Comments: A) How important are the different techniques (NMR, LC-MS, etc.) used in deciphering the various findings?  B) The majority of the studies reviewed had a small number of participants.  Does this influence the outcome of the studies?  Can the authors offer their explanation?  C) I think that the section describing the literature review should be expanded and not presented in a table format.  I view it as an important component of this review. 

- Line 93: Can the authors expand on the statement "their period has a Q10 temperature coefficient of 1"?  What is Q10?

- Line 125: Please rephrase the following: "a ~24 h cycle does not a circadian rhythm make" for better understanding.

- The authors should avoid repeating abbreviations and explanations of them in the text.  For example, in Line 258 the explanation of DLMO has already been mentioned before.  Same goes for HFD and HCD.  Please fix.

- Please use "metabolomics" instead of "metabolomic".

- Line 604: Please rewrite following sentence for better understanding: "Chua demonstrated that samples taken at the same social time presents different biological times in different individuals due to genetic differences or differences in chronotype".

- Line 730: Please replace "beenare" with "been.

- Figures S1 and S2 are not mentioned in the main text.  Please fix.

Author Response

Please see the attachment for author responses. 
